# Effects of Different Fish Diets on the Water Quality in Semi-Intensive Common Carp (*Cyprinus carpio*) Farming

László Berzi-Nagy [1,*], Attila Mozsár [1], Flórián Tóth [1], Dénes Gál [1], Zoltán Nagy [1], Sándor Alex Nagy [2], Éva Kerepeczki [1], László Antal [2,†] and Zsuzsanna J. Sándor [1,†]

1   Institute of Aquaculture and Environmental Safety Research Centre for Aquaculture and Fisheries, Hungarian University of Agriculture and Life Sciences, 5540 Szarvas, Hungary; mozsar.attila@uni-mate.hu (A.M.); toth.florian@uni-mate.hu (F.T.); gal.denes.73@gmail.com (D.G.); nagy.zoltan84@uni-mate.hu (Z.N.); eva.kerepeczki@gmail.com (É.K.); jakabne.sandor.zsuzsanna@uni-mate.hu (Z.J.S)
2   Department of Hydrobiology, University of Debrecen, 4032 Debrecen, Hungary; nagy.sandor.alex@science.unideb.hu (S.A.N.); antal.laszlo@science.unideb.hu (L.A.)
*   Correspondence: lachus160@gmail.com
†   These authors contributed equally.

**Abstract:** Semi-intensive common carp (*Cyprinus carpio*) farm technology uses several feed types affecting the growth performance; however, we know less about their long-term effects on water quality. Herein, we evaluated the effects of three commonly used feeds—moderate levels of fish meal and fish oil feed (FF), plant meal and plant oil feed (PF), and cereal feed (CF) on the nutrient (total nitrogen (TN), total phosphorus (TP) and organic matter (OM)) content of the pond water. The experiment was carried out over three consecutive years from juveniles to market-sized fish. The type of feed affected the net yields, but generally, it did not affect the water quality. The year of sampling, however, was a significant factor affecting TN, TP, and OM, whose concentrations decreased during the three years. Our findings highlight that the age of the stocked fish on water quality has a more pronounced effect than the nutrient profile of the supplementary feed. Additionally, the plant-based feed could provide comparable net yields as the fish meal-based feed without additional nutrient loading in the water column, reinforcing the sustainability of alternative feeds in semi-intensive carp farming.

**Keywords:** aquafeed; water quality; nitrogen; phosphorus; *Cyprinus carpio*

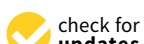



## 1. Introduction

As the human population grows and the demand for aquaculture products increases, the intensification of aquaculture practices [1] and increase in the number of farms are inevitable, both aggravating the pressure on the natural environment in the form of enhanced exploitation of resources and potential organic and inorganic pollutants. The nutrient content and composition of the diet are the main factors affecting nutrient loading by fish [2], thus, the dietary practices drive the extent of environmental effect. The combined goals of intensive yields and low pollution potential can be achieved through the use of high-quality feeds, which have good digestive properties and thus are better utilized, resulting in fewer waste nutrients per unit yield [3–7]. This effort created different feed types; the most frequently used ingredients are different terrestrial plants or industrial by-products, based on the cheapest local sources. However, the use of animal or plant-based proteins and lipids and their effects need to be carefully designed on a case-by-case basis for every species [8]. Besides their usefulness for growth, the environmental risk of novel feed types in comparison with traditional cereal feeds or fish meal-based diets is also important to estimate before wide-scale use of the given feed type is encouraged.

Common carp (*Cyprinus carpio*) has a dominant role in Central and Eastern European aquaculture [9]. The most associated production technology with this species is

semi-intensive or extensive. The semi-intensive technology uses mainly cereals for supplementary feeding and fertilization (i.e., manuring) to enhance the natural productivity of the pond [10,11]. In this technology, the production period lasts three consecutive years, the ponds are filled in spring (March–April) and are drained and harvested in autumn (September–October). The ponds are left dry in the winter allowing the aeration of the pond sediment, removing anaerobic bacteria and harmful substances, such as ammonia. At the end of the 20th century, an intensification trend and switch from cereal feeds to artificial fish meal-based feeds [12] and plant-based feeds [13] were observed in carp farming experiments.

Here we aimed to assess the effect of different supplementary feed types on production indices (i.e., survival, specific growth rate, feed conversion ratio, individual weight gain, and net yield) and water quality (i.e., total nitrogen, total phosphorus, and organic material). We designed an outdoor pond experiment to monitor the whole (3 years long) production cycle of semi-intensive common carp using three different diets: fish meal/fish oil containing feed (FF), plant-based feed (PF), and cereal feed (CF). Based on the higher quality of artificial feed [14], we expected better production indices in the ponds fed with FF and PF and lower concentrations in the water quality parameters than the ponds fed with CF. Additionally, the substitution of fish meal and fish oil with plant-based nutrients in PF feed can result in lower digestibility [15] and increase the level of antinutritional factors [16–18]. Consequently, we presumed that decreased growth performance and excessive nutrient and organic matter loadings will appear in the PF ponds compared to the FF ponds.

## 2. Materials and Methods

The 3 year-long study with three diet types using common carp monoculture was completed between 2013 and 2015 at the Research Centre for Aquaculture and Fisheries, Szarvas, Hungary. Two identical earthen ponds (average area: 1772 m$^2$, depth: 1.3 m) were assigned to each diet type. The water demand of ponds was supplied by an oxbow lake of River Körös. To follow the semi-intensive fish farming protocols, the individuals were introduced to the system in spring and harvested in autumn. In winter, the fish were held in wintering ponds, where they stopped feeding and entered a dormant phase. The same fish stock was used during the whole experiment; thus, they were stocked as fingerlings in 2013 and harvested as market-sized carps in 2015. On average 20,886 carps; 5285 and 1080 individuals/ha, with an average body weight of 0.68 g; 61.46 g and 748 g were introduced into the lakes in the years of 2013, 2014, and 2015, respectively. As the semi-intensive technology requires, cow manure was used to boost the natural production of the ponds. As the earlier age classes of common carp increasingly rely on natural food sources [19], the earlier years received manure in higher quantities. Each pond individually received 450 kg manure in 2013 and 2014, with 300 kg in 2015.

Two artificial feeds were formulated by a local aquafeed manufacturer company for rearing common carp in semi-intensive monoculture. These diets involved standard ingredients commonly used for fish feeds, were manufactured with extruding technology and they had good sinking properties, as it is common for fishpond diets in Hungary. The different feeds were specifically tailored to be used in the experiment, they were made to be isonutritious to each other and were not available in the commercial feed trade. The FF feed contained moderate levels of fish meal and fish oil, while these were replaced with a mostly soy-bean meal and linseed oil in PF feed (Table 1). These feeds were formulated to contain equal levels of crude protein and lipid (Table 2). The third diet (CF) consisted of grained winter wheat only, which is a traditional feed type used in semi-intensive carp farming in Hungary. The daily feed amount varied between 1–3.5% of MBW (metabolic body weight = BW$^{0.8}$) and was based on periodic stock samplings, where fishermen aimed to measure at least 20 individuals per pond and were conducted weekly in 2013, every three weeks in 2014 and every four weeks in 2015. In total, 1970 kg of FF feed, 2186 kg of PF feed, and 1769 kg of CF feed were distributed during the three years of the experiment.

**Table 1.** Ingredients (in percentage) of the artificial feeds (FF: feed with fish meal, PF: plant-based feed) used in the experiment, cereal feed (CF) consisted of winter wheat only.

| % | FF | | | PF | | |
|---|---|---|---|---|---|---|
| | 2013 | 2014 | 2015 | 2013 | 2014 | 2015 |
| Fishmeal 60 | 16.00 | 16.00 | 14.00 | 0 | 0 | 0 |
| Winter wheat | 8.88 | 10.08 | 20.50 | 5.60 | 8.90 | 16.50 |
| Maize | 30.00 | 30.73 | 27.50 | 29.00 | 27.00 | 27.50 |
| Full-fat soy | 6.00 | 4.03 | 6.50 | 7.80 | 9.00 | 9.50 |
| Extracted soy | 25.47 | 25.36 | 17.50 | 40.75 | 38.30 | 29.50 |
| Blood meal | 5.00 | 5.00 | 5.00 | 8.00 | 8.00 | 8.00 |
| Yeast, f.g. | 5.00 | 5.00 | 5.00 | 5.00 | 5.00 | 5.00 |
| Vit-Min mix | 2.00 | 2.00 | 2.00 | 2.00 | 2.00 | 2.00 |
| Fish oil | 1.65 | 1.80 | 2.00 | 0 | 0 | 0 |
| Linseed oil | 0 | 0 | 0 | 1.85 | 1.80 | 2.00 |

**Table 2.** Proximate composition (percentages given for wet weight) of all three feed types (FF: feed with fish meal, PF: plant-based feed, CF: cereal feed).

| % | FF | | | PF | | | CF | | |
|---|---|---|---|---|---|---|---|---|---|
| | 2013 | 2014 | 2015 | 2013 | 2014 | 2015 | 2013 | 2014 | 2015 |
| dry material | 90.95 | 91.84 | 91.86 | 90.65 | 91.71 | 92.5 | 89.32 | 96.36 | 89.8 |
| crude protein | 33.97 | 32.7 | 30.18 | 34.31 | 31.72 | 29.57 | 11.48 | 10.05 | 7.12 |
| crude fat | 6.21 | 6.27 | 7.38 | 5.86 | 5.92 | 7.43 | 1.18 | 1.2 | 0.24 |
| crude ash | 6.92 | 6.13 | 5.96 | 5.67 | 4.23 | 4.21 | 1.68 | 8.24 | 2.11 |
| nitrogen | 5.44 | 5.23 | 4.83 | 5.49 | 5.08 | 4.73 | 1.84 | 1.61 | 1.58 |
| phosphorus | 1.02 | 1.01 | 0.01 | 0.79 | 0.73 | 0.01 | 0.40 | 0.40 | 0.00 |
| organic matter | 0.84 | 0.86 | 0.86 | 0.85 | 0.87 | 0.88 | 0.88 | 0.88 | 0.88 |

The production parameters of fish were calculated based on the following equations:

1. Survival (Survival %) = 100 × (number of fish at harvest) × (number of fish at stocking)$^{-1}$
2. Specific growth rate (SGR % day$^{-1}$) = 100 × ln (average body weight at harvest × (average body weight at stocking)$^{-1}$) × (days)$^{-1}$
3. Feed conversion ratio (FCR) = (feed distributed) × (biomass weight gain)$^{-1}$
4. Weight gain (WG g fish$^{-1}$) = average body weight at harvest (g) − average body weight at stocking (g)
5. Net yield (NY kg/ha) = (weight of biomass at harvest (kg) − weight of biomass at stocking (kg)) × pond area$^{-1}$ (ha)

Integrated water samples from the whole water column were collected weekly in 2013, every two weeks in 2014 and monthly in 2015. The total nitrogen (TN) and the total phosphorus (TP) measurements used peroxodisulfate for digestion and the liberated forms were measured using a spectrophotometer [20], while the organic material (OM) was estimated from the concentration of the volatile suspended solids (VSS), measured by the weight loss-on-ignition after membrane filtering [21]. These measurements were conducted according to the guidelines of the Hungarian Standards Institution [22–24], for total nitrogen, total phosphorus and volatile suspended solids, respectively.

The effects of feed types, the different years, and the effects of feed types within the specific years on water quality indices were analyzed using Kruskal–Wallis H tests. Subsequent pairwise comparisons were made on the levels of significant factors. The statistical analyses were performed in the IBM SPSS Statistics software package [25], and significance levels were determined at $p < 0.05$.

## 3. Results

Our results of production indices support that the artificial feed types (FF, PF) performed generally similar to each other but both were better compared to the cereal feed (CF). The difference between the artificial feed types and the control feed was the most conspicuous in the yearly weight gain (WG) and net yield (NY) indices (Table 3). In the second year of the experiment, we observed the lowest survival rates for all groups, especially for the PF group. The exact reasons for the lower survival were unknown, and as the mortality was assessed only at the end of the season, the surviving individuals received relatively more feed in this group and showed somewhat higher individual weight gain in the period.

**Table 3.** Yearly production parameters (mean $\pm$ SD) by treatments (FF: feed with fish meal, PF: plant-based feed, CF: cereal feed).

| | | 2013 | 2014 | 2015 |
|---|---|---|---|---|
| FF | Survival (%) | 78.6 $\pm$ 3 | 72.7 $\pm$ 11.2 | 89.5 $\pm$ 1.5 |
| | SGR (% day$^{-1}$) | 3.37 $\pm$ 0.05 | 1.06 $\pm$ 0.08 | 0.66 $\pm$ 0.03 |
| | FCR | 1.63 $\pm$ 0.11 | 2.51 $\pm$ 0.23 | 2.5 $\pm$ 0.1 |
| | WG (g fish$^{-1}$) | 76.6 $\pm$ 4.9 | 641.5 $\pm$ 81.8 | 1686.1 $\pm$ 137.5 |
| | NY (kg ha$^{-1}$) | 1219 $\pm$ 39.1 | 2034.8 $\pm$ 129.4 | 1669.5 $\pm$ 88.9 |
| PF | Survival (%) | 75 $\pm$ 2 | 49.2 $\pm$ 6.5 | 92.3 $\pm$ 0.2 |
| | SGR (% day$^{-1}$) | 3.33 $\pm$ 0.12 | 1.23 $\pm$ 0.1 | 0.6 $\pm$ 0.01 |
| | FCR | 1.91 $\pm$ 0.08 | 3.54 $\pm$ 0.01 | 2.59 $\pm$ 0.01 |
| | WG (g fish$^{-1}$) | 60.9 $\pm$ 10.1 | 863 $\pm$ 199.4 | 1735.2 $\pm$ 48.6 |
| | NY (kg ha$^{-1}$) | 916.2 $\pm$ 202.2 | 1614.5 $\pm$ 107.5 | 1778.5 $\pm$ 63 |
| CF | Survival (%) | 68.3 $\pm$ 2.4 | 57.8 $\pm$ 9 | 91.9 $\pm$ 0.9 |
| | SGR (% day$^{-1}$) | 3.23 $\pm$ 0.08 | 1.12 $\pm$ 0.12 | 0.57 $\pm$ 0.01 |
| | FCR | 2.12 $\pm$ 0.22 | 3.31 $\pm$ 0.22 | 3.04 $\pm$ 0.09 |
| | WG (g fish$^{-1}$) | 52.7 $\pm$ 5.6 | 609.3 $\pm$ 177.1 | 1282.9 $\pm$ 65.3 |
| | NY (kg ha$^{-1}$) | 732.1 $\pm$ 24 | 1418.5 $\pm$ 122.6 | 1321.4 $\pm$ 66.5 |

Based on the test results of homogeneity of variance (Levene's test, $p < 0.01$), the Kruskal–Wallis test was chosen to analyze differences between groups. Considering water quality parameters, PF ponds had slightly higher average TN (1.38 mg L$^{-1}$), TP (0.16 mg L$^{-1}$), and organic material expressed in VSS (19.72 mg L$^{-1}$) concentrations compared to the ponds with the other feed types (1.18 mg L$^{-1}$, 0.15 mg L$^{-1}$, 17.28 mg L$^{-1}$ for FF ponds and 1.21 mg L$^{-1}$, 0.15 mg L$^{-1}$, 17.20 mg L$^{-1}$ for CF ponds respectively), however, this difference was not consistent within the specific years. The highest TN values were registered in FF ponds in 2013 (FF: 1.54 mg L$^{-1}$; PF: 1.48 mg L$^{-1}$; CF: 1.49 mg L$^{-1}$), and in PF ponds in 2014 (FF: 1.04 mg L$^{-1}$; PF: 1.45 mg L$^{-1}$; CF: 1.09 mg L$^{-1}$), and 2015 (FF: 0.77 mg L$^{-1}$; PF: 0.95 mg L$^{-1}$; CF: 0.61 mg L$^{-1}$). The highest TP concentrations were in CF ponds in 2013 (FF: 0.198 mg L$^{-1}$; PF: 0.185 mg L$^{-1}$; CF: 0.203 mg L$^{-1}$), and in PF ponds in 2014 (FF: 0.126 mg L$^{-1}$; PF: 0.153 mg L$^{-1}$; CF: 0.118 mg L$^{-1}$) and 2015 (FF: 0.101 mg L$^{-1}$; PF: 0.118 mg L$^{-1}$; CF: 0.092 mg L$^{-1}$). The highest organic material (i.e., VSS) values were in PF ponds in 2013 (FF: 22.45 mg L$^{-1}$; PF: 26.11 mg L$^{-1}$; CF: 22.56 mg L$^{-1}$) and in 2014 (FF: 13.24 mg L$^{-1}$; PF: 14.39 mg L$^{-1}$; CF: 11.48 mg L$^{-1}$) and in CF ponds in 2015 (FF: 13.1 mg L$^{-1}$; PF: 17.24 mg L$^{-1}$; CF: 17.37 mg L$^{-1}$). Ultimately, the differences in water quality parameters were not statistically significant compared between the three feed types. Testing the feed types separately for the given three years, however, showed a continuous separating trend between the water quality of the given feeds, which were reflected by decreasing p-values. Only in the third year (2015) was TN significantly ($\chi^2(2) = 8688$, $p = 0.013$) different between feed types. Pairwise comparisons showed that this was the result of PF ponds having higher TN than CF fed ponds in 2015. Additionally, the measured water quality variables indicated significant temporal (yearly) variance and decreasing concentrations during the consecutive years of the experiment (Figure 1). Analyzing the

year as a grouping variable, and the Kruskal–Wallis test showed it to be highly significant for all water quality variables (TN: $\chi^2(2) = 34,516$, $p < 0.001$, TP: $\chi^2(2) = 73,769$, $p < 0.001$, VSS: $\chi^2(2) = 19,744$, $p < 0.001$) (Figure 2).

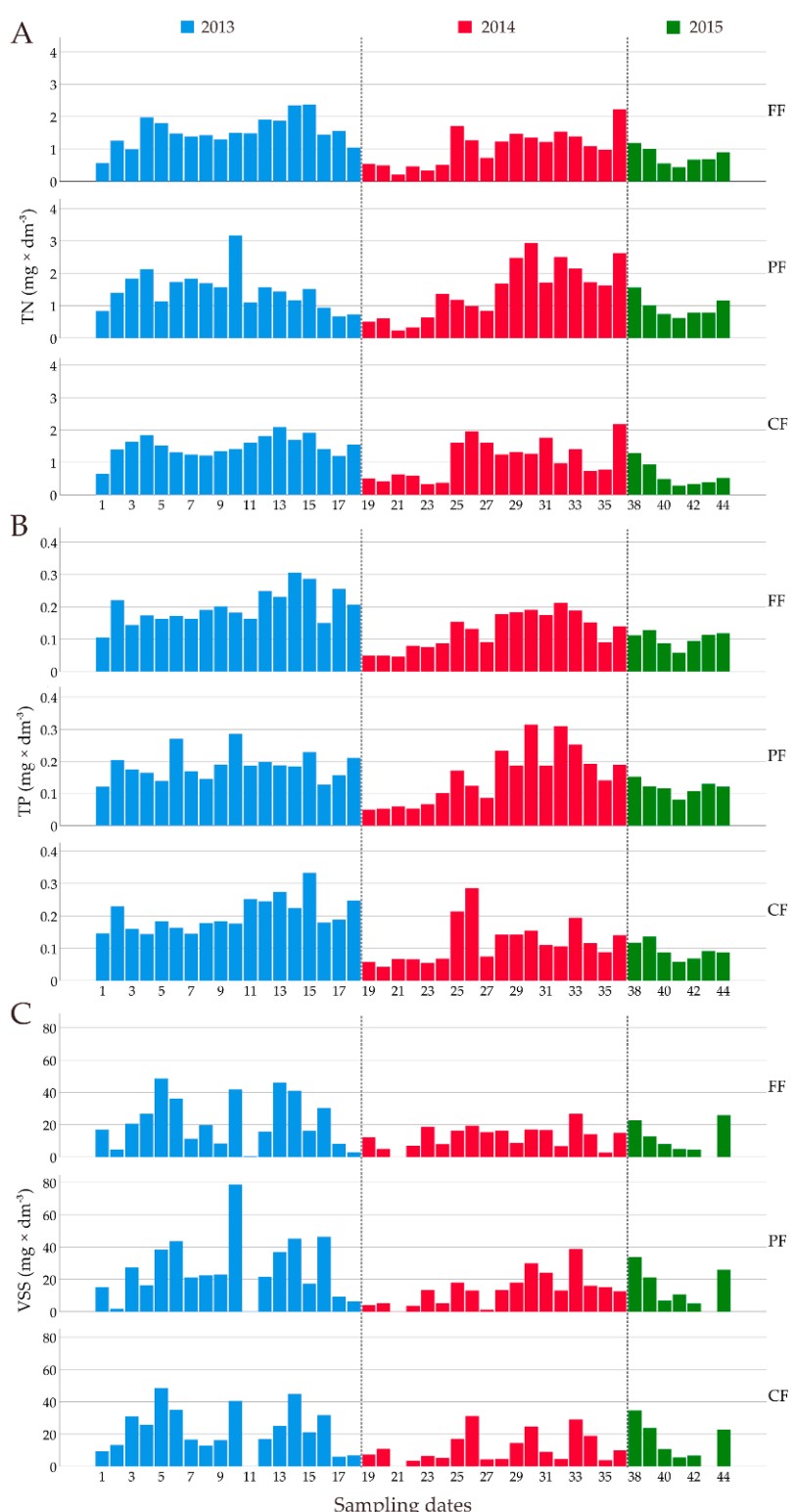

**Figure 1.** Annual profile of the three water quality variables (TN: total nitrogen, TP: total phosphorus, VSS: volatile suspended solids) in case of all three feed types (FF: feed with fish meal, PF: plant-based feed, CF: cereal feed). Figures were generated by the IBM SPSS software package (version 25).

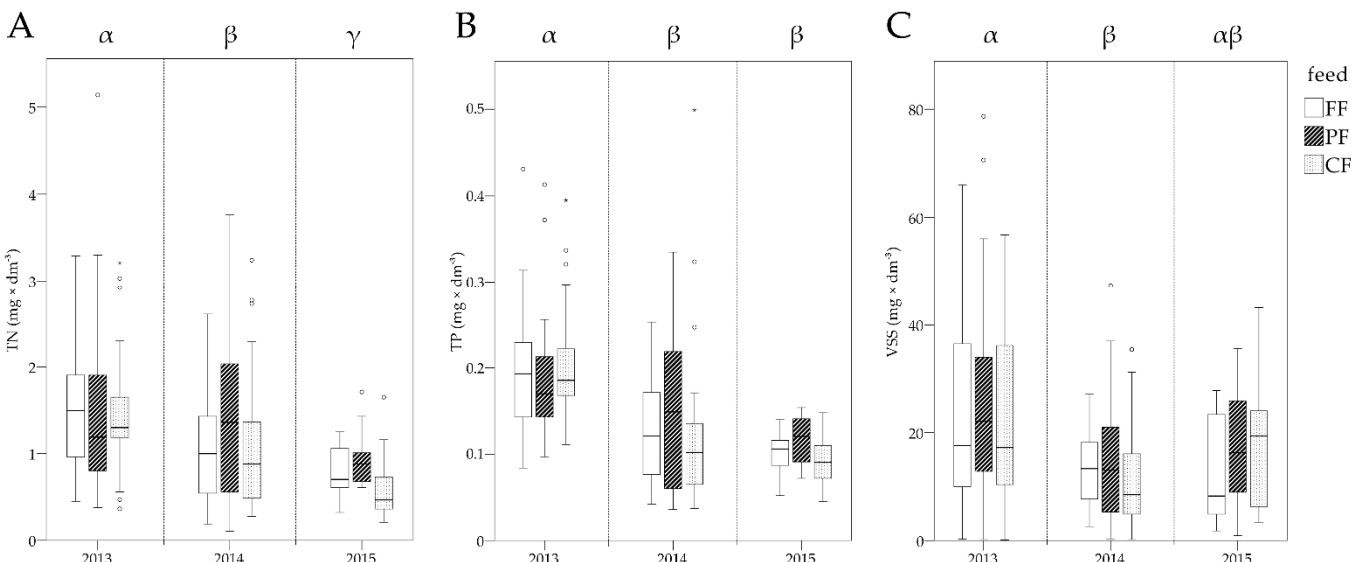

**Figure 2.** Water quality changes between years and treatments (FF: feed with fish meal, PF: plant-based feed, CF: cereal feed), indicated by total nitrogen (**A**), total phosphorus (**B**), and organic material (measured as volatile suspended solids—VSS) (**C**) concentrations. The different Greek letters (α, β, γ) represent statistically significant ($p < 0.05$) differences between the years for the given water quality variable. Tests and figures were generated by the IBM SPSS software package (version 25).

Concluding the results provided by the Kruskal–Wallis tests, the year as a factor significantly affected all water quality parameters, but the feed types resulted in only one difference in TN, which was significant in the last year only. This shows the temporal changes in water quality to be dominant over the applied feed types in our experiment. These temporal changes were likely the results of specific effects coming from the characteristic of the weather and the stocked fish in any given year.

## 4. Discussion

Feeding in aquaculture contributes directly and indirectly to the water quality and ultimately fish welfare as well. As the concentrations of nutrients increase in water, the risk of their potentially harmful solutes in combination with the environmental conditions affects survival and needs to be optimized [26]. Intensification in pond farming demands qualitative changes in feeds and necessitate fish feeds with higher protein content compared to traditional cereal feeds. The most frequent sources of protein in fish feeds are still fish meal and soybean meal. Although the nutritional profile is similar in these artificial diets, their utilization and digestibility efficiency differ, and thus different contributions in water quality can be expected. The present study analyses the full production cycle (i.e., from juvenile to market sized) of common carp and, in this regard, it provides multi-year data to elucidate changes in water quality under semi-intensive pond farming. In our experiment, the year of production was a significant predictor of water quality, however, the type of feed was generally not. The annual changes highlight that the role of fish in environmental nutrient dynamics undergoes substantial changes during ontogeny.

The higher net yields and weight gain in artificial (FF, PF) feeds compared to the cereal (CF) diets indicated the effectiveness of intensification. The higher protein content of FF and PF diets (>30% crude protein) was closer to the nutritional requirements of the carp [27,28]. This compliance to the protein requirements resulted in better FCR and growth rate and consequently higher net yield for artificial feed types. In ponds fed with CF, the protein requirements of the fish could only be temporarily satisfied when zooplankton organisms were in abundance, however, when planktonic biomass seasonally decreased, the lack of protein slowed the growth of fish. Having continuous access to protein sources and balanced amino acid profiles contributed to the unhindered growth of fish in FF and PF ponds, especially in the second and third years. This result may suggest that utilization of

high-protein feed types might be most beneficial in the older (1- or 2-year-old) age groups of carp under semi-intensive farming.

Although the formulated diets (FF, PF) were designed to be consumed more efficiently than cereal feeds, these also contained higher N and P concentrations, and these opposing effects seemingly extinguished each other, as there was only one statistically significant difference in the water quality of ponds fed with different feed types during the three-year period of the experiment, which occurred only in TN and only in the third year. The lack of differences disproves our initial notion that the high-quality feed types provide better water quality, but it reinforces that high-protein feed types are adoptable substitutes for the traditional cereal-based diets to increase yield without adverse environmental effects in the ponds under semi-intensive technology. Furthermore, as stagnant ponds can be characterized with relatively quick nutrient mineralization and turnover rate, enhancing natural processes removing natural pollutants, like phosphorus from water [29], such environmental factors may also contribute to the lack of significant differences in water quality between the different feed types in our experiment. It should also be noted that the different diets may not always alter water quality parameters substantially [30], but in certain cases, differences in organic matter and chlorophyll-*a* [31] or conductivity [32] were reported in similar experiments.

We also aimed to describe differences in the water quality between FF and PF diets. As the PF diet did not affect the measured nutrient concentrations in the water compared to FF, it supported the idea that PF feeds can potentially fully substitute FF feeds without having negative effects on water quality in the semi-intensive common carp production technology. Alternative protein sources substituting for fish meal and fish oil can support a sustainable future for the expanding aquaculture sector [33].

Lastly, the difference in nutrient concentrations of water columns between years could rise from the considerable allometric changes in the role of fish in nutrient dynamics [34,35]. The mass-specific excretion rate is higher in juvenile fish in comparison with adults [35, 36], accordingly, the amount of excreted nitrogen and phosphorus per unit biomass was higher in the first year. Additionally, zooplankton consumption is more emphasized in young carps [37], resulting in less grazing pressure on phytoplankton. The increased phytoplankton density accelerated the internal nutrient cycling and kept the nutrients in the water column [38]. Carps are omnivorous fish species, which reportedly have the potential to increase nutrients in the water column even at earlier stages of development. In mesocosm experiments with *Carassius auratus* fingerlings, Huang et al. [39] reported higher concentrations of TN, TP, and TSS. In the following years, the mass-specific excretion rate decreased, and the carp individuals switched from zooplanktivory to benthivory [40]. The former phenomenon decreased the direct contribution of fish to the internal nutrient loading by excretion, while the latter increased zooplankton density and thus also the grazing pressure on phytoplankton. The benthivory, furthermore, increased the turbidity due to the stirring effect exerted along with bioturbation [41] and could adversely affect phytoplankton growth. Although the bioturbation can contribute considerably to the internal nutrient (especially phosphorus) loading, providing a nutrient flux from the bottom to water column [42,43], it did not result in an increment in nutrient concentration in this study. On the one hand, the factors discussed above presumably suppressed this nutrient flux. On the other hand, the well-oxygenized environment in the ponds did not favor the release and ratio of orthophosphate [44]. Substantial differences in the role of fish in nutrient dynamics occur between juvenile and adult life stages [36]. In accordance with this, the first year provided more significant differences in water quality compared to the following years. The role of fish in nutrient dynamics was expressed via numerous pathways, while several of these influence the internal nutrient dynamics in the same direction, others affect inversely [34,45].

Our findings showed that the date of sampling is a more powerful predictor compared to the added feed type and support the findings of other authors [14]. However, the amount of added feed and the level of intensification can potentially affect the relationship between

the feed types and the water quality [46–50]. In order to explain the phenomenon in more detail, further research on carrying capacity and retention dynamics of the fishponds could be highly valuable. Additionally, the nutritional pathways could be described more accurately if further insight into the nutrient concentrations of sediment, or planktonic and benthic biomass calculations are provided. Regardless of the exact underlying mechanism, and the lower survival rates observed in the second year of the experiment, our results support that the highly nutritious feed types can safely increase the yield of semi-intensive pond aquaculture without producing adverse effects on water quality variables. Considering only the net feed costs in our experiment (FF: 0.7 €/kg, PF: 0.5 €/kg and CF: 0.1 €/KG) and assuming the wholesale net selling price of live carp to be 2.5 €/kg [51], we can calculate the profit in our case to be 11,000 € with FF, 9730 € with PF and 8437 € with CF through the three years of production. In this case of intensification, the higher quality feed types could have paid off by the higher carp yield they generated compared to the control feed. We need to emphasize that this equation is only realistic if the market demand for additional fish products is realized for the farmer and the relevant operational costs (e.g., transporting, storing, and handling) do not increase with the intensification. Based on our study, the change to alternative, higher quality feed types does not necessarily increase the expenditure to keep stable water quality in semi-intensive pond farming. Although the traditional cereal-based diets are the most used feeds in Hungarian inland pond farming, the farmers need to regularly consider their options for intensification in the changing European market. The interchangeability of high-protein feed ingredients was also an important outcome in our experiment, which indicates that fishmeal and fish oil dependence can be reduced without detrimental changes in yield or water quality. The substitution of limited feed ingredients, like fish meal, is an essential endeavor for the sustainability of the ever-expanding pond aquaculture [52]. Furthermore, as the National Fisheries Strategic Plan of Hungary emphasizes the role of fishponds to provide stock for natural waters [53], the trend of sustainable and intensified inland aquaculture in Hungary could provide additional benefits for national fisheries as well.

**Author Contributions:** Conceptualization, L.A. and É.K.; formal analysis, L.B.-N.; methodology, L.B.-N. and S.A.N.; software, A.M.; investigation, D.G. and Z.N.; data curation, F.T.; writing—review and editing, L.B.-N., Z.J.S. and F.T.; visualization, L.A.; supervision, S.A.N.; project administration, Z.J.S.; funding acquisition, Z.J.S. and L.A. All authors have read and agreed to the published version of the manuscript.

**Funding:** This research was funded by the ARRAINA project (Advanced Research Initiatives for Nutrition and Aquaculture, project N°288925-EU FP7). The views expressed in this work are the sole responsibility of the authors and do not necessarily reflect the views of the European Commission.

**Institutional Review Board Statement:** All applicable international, national, and/or institutional guidelines for the care and use of animals were followed by the authors. The Institute is licensed to carry out animal experiments by the local animal health authority of the Békés county (Békéscsaba, Hungary; registration number: 1/2002).

**Informed Consent Statement:** Not applicable.

**Data Availability Statement:** The data that support the findings of this study are available from the corresponding author upon reasonable request.

**Acknowledgments:** We are also thankful for the work delivered by the many colleagues of the Research Institute for Fisheries and Aquaculture, Szarvas and by Aranykárász LP, as partner in the project. Writing the article was supported by project no. TKP2020-NKA-24 from the National Research, Development and Innovation Fund of Hungary, financed under the 2020-4.1.1-TKP2020 funding scheme.

**Conflicts of Interest:** The authors declare no conflict of interest.

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
