# Peer review of "Effects of Different Fish Diets on the Water Quality in Semi-Intensive Common Carp (Cyprinus carpio) Farming"

_water, doi:10.3390/w13091215_

Round 1

Reviewer 1 Report

The authors monitor the effect of three feeds on nutrient content (TP, TN, OM) in the pond water of semi-intensive production systems. Considering the role of carp farming in Central and Eastern Europe and the drainage of fish ponds to natural rivers and lakes, this is an important subject for a broad audience. Unfortunately, phosphate species were not determined, which is important in the context of bioavailability. As a strong point of the study, feed formulation is realistic for carp farming, utilizing several plant ingredients in contrast to other studies using a single ingredient, which provokes early antinutritional and malnutritional effects. Making use of such a balanced diet may in part explain that no mayor differences in yield were observed.

The manuscript is well prepared and easy to read. The results are presented convincingly but I do suggest that the annual profile of the three nutrients (comparing the three groups) over the three year period is included.

In the discussion, you state “Regardless  of  the  exact  underlying  mechanism, our results support that the highly nutritious feed types can safely increase the yield of semi-intensive pond aquaculture without producing adverse effects on water quality variables.” Therefore, I assume differences in Tab. 3, which should be indicated. Otherwise reformulate the sentence. Also, it would be nice to provide an economic evaluation. Does the increased yield pay off for increased feed costs in PF and FF (outline a rough estimation)? This aspect should be included in the discussion. Please add a few sentences on the stocking density in the experiment with regard to the intensification of aquaculture as envisioned by the National Aquaculture Plan in Hungary, if possible. I also suggest that you provide clear recommendations for the farmer at the end of the discussion.

Author Response

  1. „The results are presented convincingly but I do suggest that the annual profile of the three nutrients (comparing the three groups) over the three year period is included.”

Response: Lines 176: We agree that the yearly variations can provide further insight on how the nutrients changed between the treatment groups. We included additional figures to present the annual profiles.

  1. „In the discussion, you state “Regardless of the exact underlying mechanism, our results support that the highly nutritious feed types can safely increase the yield of semi-intensive pond aquaculture without producing adverse effects on water quality variables.” Therefore, I assume differences in Tab. 3, which should be indicated.”

Response: Lines 139-144: We provided additional insight into the differences presented in Table 3.

Line 280: The differences in survival (Table 3) are mentioned also in the discussion.

  1. „Also, it would be nice to provide an economic evaluation. Does the increased yield pay off for increased feed costs in PF and FF (outline a rough estimation)? This aspect should be included in the discussion.”

Response: Lines 283-295: We provided an additional economic calculation. Although this is a rough estimation, as it is not the focus of the article, we concluded the effectiveness of the artificial feed types, considering the overall net yield observed during the experiment.

  1. „Please add a few sentences on the stocking density in the experiment with regard to the intensification of aquaculture as envisioned by the National Aquaculture Plan in Hungary, if possible.”

Response: Lines 79-81: We added further data on the stocking densities and individual weights applied during the experiment.

Lines 299-303: We provided additional insight about the importance of commercial fishponds in the stocking of natural waters in the National Fisheries Strategic Plan of Hungary.

  1. „I also suggest that you provide clear recommendations for the farmer at the end of the discussion.”

Response: Lines 288-295: We provided recommendations for the farmers, and factors to consider, regarding the decision of intensificating their farming operations.

Reviewer 2 Report

Some new references could be inserted.

PDF file with suggestion is included

Author Response

Reviewer #2

  1. Line 38: The reviewer suggested to add further references.

Response: We added two additional references regarding the effectiveness of aquafeeds and their implications on water quality.

  1. Line 94: The reviewer suggested to delete the sentence.

Response: Thank you. The sentence was redundant, we deleted it.

  1. Line 128: The reviewer pointed out the high mortality in 2nd year regarding the PF group and asked whether it’s the reason for the lower net yield in the given year.

Response: Indeed, in the second year there was a lower survival rate in all groups, especially in the PF group.

Lines 139-144: additional insight was provided. Unfortunately, we could not detect the exact reason for the mortality in 2014 and we did not find the carcasses. As the mortality was assessed at the end of the season by full harvesting, the mortality was not detected during the season. This could result in a relative overfeeding of the surviving individuals, which could lead to a relatively higher weight gain, but even with the higher gain, they could not compensate for the loss of individuals and ultimately the net yield was lower than in group FF (Table 3.).

  1. Lines 138-141: The reviewer asked for further details about the nutrient concentrations between the groups for the three years.

Response: Lines 155-164: We provided the yearly averages for every nutrient type, group and year.

  1. Line 183: the reviewer asked whether the sentence is logically the follow up of the previous sentence.

Response: Lines 213-214: yes, the sentence is supposed to be logically connected to the previous sentence, we included additional text for connection.

  1. Line 211: reviewer made a comment that the statement should be restricked to species level.

Response: Line 242: we agree, thank you, we added additional clarification.

  1. Line 213: reviewer asked whether the sentence is understood for the whole sector of aqauculture.

Response: Line 244: yes, the sentence was made for the whole sector, we included an additional reference as well.

  1. Line 255: the reviewer asked for further insight regarding the mortality in 2014.

Response: Line 280: the description of the mortality was detailed in the results section (Lines 139-144), in discussion we referred to it with a short text, where we assessed the efficiency of the feed types.
